# *Artemisia brevifolia* Wall. Ex DC Enhances Cefixime Susceptibility by Reforming Antimicrobial Resistance

**DOI:** 10.3390/antibiotics12101553

**Published:** 2023-10-20

**Authors:** Aroosa Zafar, Yusra Wasti, Muhammad Majid, Durdana Muntaqua, Simona Gabriela Bungau, Ihsan ul Haq

**Affiliations:** 1Department of Pharmacy, Faculty of Biological Sciences, Quaid-i-Azam University, Islamabad 45320, Pakistan; arzafar@bs.qau.edu.pk (A.Z.); yusrawasti@bs.qau.edu.pk (Y.W.); 2Cadson College of Pharmacy, Kharian 50090, Pakistan; 3Faculty of Pharmacy, Hamdard University, Islamabad 45550, Pakistan; muhammad.majid@humdard.edu.pk; 4Shifa College of Pharmaceutical Sciences, Shifa Tameer-e-Millat University, Islamabad 44000, Pakistan; msdmalik1984@gmail.com; 5Department of Pharmacy, Faculty of Medicine and Pharmacy, University of Oradea, 410028 Oradea, Romania; 6Doctoral School of Biomedical Sciences, University of Oradea, 410087 Oradea, Romania

**Keywords:** RP-HPLC, cefixime, antibacterial susceptibility testing, *Artemisia brevifolia*, traditional medicine, plant extracts, antimicrobial activity, drug resistance, phytotherapy, bioactive compounds

## Abstract

(1) Background: A possible solution to antimicrobial resistance (AMR) is synergism with plants like *Artemisia brevifolia* Wall. ex DC. (2) Methods: Phytochemical quantification of extracts (n-hexane (NH), ethyl acetate (EA), methanol (M), and aqueous (Aq)) was performed using RP-HPLC and chromogenic assays. Extracts were screened against resistant clinical isolates via disc diffusion, broth dilution, the checkerboard method, time–kill, and protein quantification assays. (3) Results: M extract had the maximum phenolic (15.98 ± 0.1 μg GAE/mgE) and flavonoid contents (9.93 ± 0.5 μg QE/mgE). RP-HPLC displayed the maximum polyphenols in the M extract. Secondary metabolite determination showed M extract to have the highest glycosides, alkaloids, and tannins. Preliminary resistance profiling indicated that selected isolates were resistant to cefixime (MIC 20–40 µg/mL). Extracts showed moderate antibacterial activity (MIC 60–100 µg/mL). The checkerboard method revealed a total synergy between EA extract and cefixime with 10-fold reductions in cefixime dose against resistant *P. aeruginosa* and MRSA. Moreover, *A. brevifolia* extracts potentiated the antibacterial effect of cefixime after 6 and 9 h. The synergistic combination was non- to slightly hemolytic and could inhibit bacterial protein in addition to cefixime disrupting the cell wall, thus making it difficult for bacteria to survive. (4) Conclusion: *A. brevifolia* in combination with cefixime has the potential to inhibit AMR.

## 1. Introduction

Antibiotic treatment is the most significant therapy for bacterial infections and has marvelously enhanced human health. At present, there are more than 14 classes of antibiotics on the market to treat several infections [1]. In the past few decades, antibiotic resistance has emerged, leading to treatment failures. Unraveling the mechanisms of resistance is a priority in order to support work to devise efficacious therapies against life-threatening resistant infections. Resistance can be due to many factors such as antibiotic inactivation by bacteria or reduced bacterial membrane penetration. Furthermore, previously susceptible bacteria can develop resistance through genetic mutations or via generating biofilms to protect bacterial colonies from exogenous damage [2]. This stems from the overuse of antibiotics, promoting evolutionary resistance in bacteria via natural selection. Healthcare professionals have stressed the need for new drugs or therapies to combat emerging resistance among pathogens to antibiotics already available [3]. Therefore, there is ongoing research using both herbal and synthetic compounds in an attempt to find effective therapies against resistant bacterial isolates. Medicinal plants with proven antibacterial activity are good options, considering their wide availability and safety profiles [4].

The traditional medicinal system has established its role as a therapeutic alternative to the allopathic system to treat wounds, abrasions, and infections [5] and has been used and developed through the centuries since ancient times [6]. This natural-compound-based medicine has contributed effective antimalarial (artemisinin and quinine) and anticancer drugs (vinblastine and taxols) to treat the respective diseases [7]. Accordingly, scientists are inclined to investigate medicinal plants as an alternative source of antibiotics. Plants are enriched with several secondary metabolites, which exhibit antimicrobial properties. For example, terpenoids and essential oils have shown antibacterial effects by affecting cell membrane permeability [8].

Essential oils, besides other numerous applications [9], can be used in synergy with antibiotics to alter the bacterial membrane permeability [10]. Furthermore, tannins bind to protein, alkaloids intercalate into DNA, and flavonoids bind in an adhesion complex with the cell and also inhibit the enzymatic activity of bacterial cells [11]. Furthermore, tannins demonstrate dose-dependent antibacterial effects by potentially inhibiting extracellular enzymes, modulating bacterial cell metabolism via oxidative phosphorylation inhibition, limiting the substrate required for bacterial growth, and targeting proteolytic enzymes that inhibit protein synthesis [12].

Pakistan’s flora includes 5521 species representing 1572 genera with ethnomedicinal significance [13]. Notable among them, *Matricaria chamomilla* combats *S. aureus*, *M. tuberculosis*, and Helminthes, while plants like *Allium sativum*, *Piper betel*, *Malus sylyestris*, *Pimenta dioica*, and *Syzygium aromaticum* exhibit antibacterial properties, *Panax notoginseng* targets *E. coli*, and *Aloe barbadensis* and *Lawsonia inermis* are potent against *S. aureus*. Moreover, *Medicago sativa* and *Erythroxylum coca* combat active Gram-positive and Gram-negative bacteria, respectively [11].

*Artemisia* L. is a common and diverse genus of the Asteraceae family consisting of more than 500 species with significant therapeutic and economic importance [14]. Currently, 38 species of *Artemisia* have been identified and botanically reported in Pakistan, which are mainly present in dry and semidry areas of Khyber Pakhtunkhwa, Northern Punjab, Baluchistan, and Kashmir [14]. *Artemisia brevifolia* Wall. ex DC is locally known as “Tarkha” or “Mori”. It is one of the most commonly found species in cold desert areas, the Himalayas, Ladakh, Kashmir, and Afghanistan [15]. It is broadly spread in many areas of Pakistan over 2500 m in altitude, including Chitral, Gilgit, Swat, Baltistan, Khaghan, Astor, and the Deosai plains [16]. Traditionally, *A. brevifolia* has been used as an antiseptic, anthelmintic, antidiabetic, carminative, blood purifier, stress reliever, diuresis, pain reliever, antitussive, stomachic, febrifuge, and as an antidote for the scorpion sting [15].

Alkaloids, terpenoids, essential oils, and flavonoids have proven antibacterial activity. Previous studies on *A. brevifolia* validated the presence of alkaloids, flavonoids, terpenoids, glycosides, essential oils, and vitamins. It is hypothesized that *A. brevifolia* possesses significant antibacterial proficiency [16,17,18] owing to the presence of formerly reported chemical constituents.

Therefore, this research unveils a novel dimension in the scientific exploration of a plant species belonging to the commercially significant genus Artemisia, known for its well-established ethno-medicinal applications. Remarkably, despite its established reputation, the comprehensive exploration of the antimicrobial potential of this particular plant species remains conspicuously underrepresented in the existing scientific literature. An innovative approach to countering the formidable challenge of drug resistance is emerging: the synergistic interplay of phytochemicals with contemporary antibiotics. In this pioneering study, we delve into uncharted territory by thoroughly assessing the hitherto unexplored synergistic efficacy of *A. brevifolia*. This marks the very first instance in which such an evaluation has been undertaken, thereby adding a novel dimension to the field of antimicrobial research. Furthermore, in an unprecedented stride, our research embarks upon a biocompatibility study of *A. brevifolia*, a facet that has hitherto remained unreported in scientific investigations. This groundbreaking endeavor underscores our commitment to shedding new light on the properties and applications of this plant species, further emphasizing the unique and pioneering nature of our research.

In sum, this research aimed to study the antibacterial effect of *A. brevifolia* in resistant clinical isolates as a single therapy and as a combination therapy with an antibiotic. Here, we demonstrate the synergistic activity of *A. brevifolia* extracts and cefixime in reducing the growth of cefixime-resistant clinical isolates. Our results provide evidence of the antibacterial activity of *A. brevifolia* and present it as a source of isolating antibacterial compounds.

## 2. Results

### 2.1. Percentage Yield

*A. brevifolia* crude extracts were prepared in four solvents spanning the polar to non-polar range. There was the highest percent extract recovery in aqueous (Aq) extract (6.1% *w*/*w*), which gradually declined with decreasing solvent polarity. The M, EA, and NH solvents extracted the phytoconstituents from *A. brevifolia* with values of 4.65, 2.95, and 1.65%, respectively, of the total weight of dry plant (12 kg) used for extraction.

### 2.2. Phytochemical Analysis

#### 2.2.1. Total Flavonoid Content Estimation

The total flavonoid content in *A. brevifolia* extracts is presented in Table 1. It was calculated with the calibration curve, y = 0.0649, x − 0.043, R^2^ = 0.9927. The results show that the highest TFC content was present in the M extract, followed by the Aq extract and then the EA extract. The lowest number of flavonoids were present in the NH extract.

#### 2.2.2. Total Phenolic Content Estimation

Total phenolic content was expressed as µg GAE/mgE and calculated with the calibration curve, y = 0.0915, x − 0.098, R^2^ = 0.9939 (Table 1). The highest phenolic content was found in the M extract, followed by the Aq extract and then the EA extract, while the NH extract had the lowest phenolic content.

#### 2.2.3. Secondary Metabolite Estimation

*A. brevifolia* extracts were qualitatively evaluated for the presence of phytochemicals (Table 2). The results showed that cardiac glycosides, alkaloids, and terpenoids were present in the NH extract. The EA extract had cardiac glycosides, anthraquinone glycosides, and terpenoids. On the contrary, the M and Aq extracts exhibited glycosides (cardiac, anthraquinone, and coumarin), alkaloids, and tannins. Saponin content was observed only in the Aq extract.

#### 2.2.4. RP-HPLC Analysis: Detection of Polyphenolic Content

The quantification of various polyphenols was performed using the RP-HPLC-DAD method by comparing the UV spectra and retention times of the standard with those of test extracts (Table 3 and Figure 1 and Figure 2). Polyphenols in four extracts of *A. brevifolia* were quantified using 14 standards. The NH extract showed the maximum concentration of catechin (0.82 ± 0.09 µg/mgE) while all other polyphenols were present in minute quantities. The EA extract was found to have maximum concentrations of apigenin (8.65 ± 0.012 µg/mgE) and syringic acid (2.54 ± 0.04 µg/mgE). Apigenin (3.13 ± 0.015 µg/mgE), rutin (2.85 ± 0.025 µg/mgE), and catechin (2.43 ± 0.08 µg/mgE) were detected at the highest concentrations in M extract among all other extracts, whereas Aq extract showed all polyphenols had rutin (1.11 ± 0.04 µg/mgE) and catechin (1.35 ± 0.08 µg/mgE) in the maximum concentrations, though those were much lower amounts than for the other extracts. As depicted in the results, more polyphenols were quantified in the M extract than in the EA extract, suggesting it was the best candidate for bioactivity evaluation.

### 2.3. Clinical Bacterial Isolates Were Resistant to Cefixime

Susceptibility of selected clinical isolates of Gram-positive (MRSA and *S. hemolyticus*) and Gram-negative (*E. coli* and *P. aeruginosa*) bacteria to antibiotics was assessed using the disc diffusion method (Table 4). According to the Clinical and Laboratory Standard Institute (CLSI) guidelines, any antibiotic with ZOI ≤ 14 mm is considered resistant at the standard CLSI set dose [19]. The results showed that the growth of selected clinical isolates was inhibited by ciprofloxacin, doxycycline, clarithromycin, and lincomycin. Their ZOI ranged from 17 to 30 mm, 16 to 35 mm, 24 to 37 mm, and 20 to 35 mm for *E. coli*, *P. aeruginosa*, *S. hemolyticus,* and MRSA, respectively. Interestingly, all four clinical isolates were resistant to cefixime (10 µg) with no ZOI values. This corresponds to the CLSI guidelines that set the resistance ZOI value for cefixime at ≤15 mm at 5 µg/disc [19]. Hence, cefixime was used in combination with *A. brevifolia* extracts for synergistic studies against cefixime-resistant clinical bacterial isolates.

**Table 3 antibiotics-12-01553-t003:** RP-HPLC-DAD analysis of *A. brevifolia* extracts for their polyphenolic composition.

Polyphenol	LOD	LOQ	RT	Λ [20]	Polyphenols (µg/mg of Sample)			
					NH	EA	M	Aq
Vanillic Acid	0.02	0.06	11.610	257	-	-	0.21 ± 0.5 ^h^	0.13 ± 0.04 ^f^
Rutin	0.18	0.54	15.278	257	-	0.55 ± 0.5 ^c^	2.85 ± 0.025 ^b^	1.11 ± 0.04 ^b^
Gallic Acid	0.05	0.16	4.166	279	0.16 ± 0.3 ^d^	0.33 ± 0.3 ^e^	0.00 ± 0.01	0.156 ± 0.02 ^f^
Catechin	0.04	0.11	9.011	279	0.82 ± 0.09 ^a^	-	2.43 ± 0.08 ^c^	1.35 ± 0.08 ^a^
Syringic Acid	0.05	0.14	12.188	279	0.24 ± 0.08 ^c^	2.54 ± 0.04 ^b^	1.22 ± 0.09 ^e^	0.28 ± 0.09 ^d^
Apocynin	0.01	0.04	14.405	279	0.04 ± 0.04 ^f^	0.08 ± 0.05 ^g^	-	0.07 ± 0.05 ^g^
Coumaric Acid	0.08	0.23	16.949	279	0.26 ± 0.03 ^c^	0.42 ± 0.01 ^d^	1.19 ± 0.09 ^e^	0.23 ± 0.04 ^e^
Gentisic Acid	0.06	0.18	8.464	325	-	0.00 ± 0.01	0.43 ± 0.07 ^g^	0.90 ± 0.01 ^c^
Caffeic Acid	0.03	0.09	9.970	325	0.10 ± 0.05 ^e^	0.21 ± 0.01 ^f^	0.44 ± 0.02 ^g^	0.22 ± 0.015 ^e^
Ferulic Acid	0.03	0.10	13.624	325	0.14 ± 0.05 ^de^	-	-	-
Luteolin	0.25	0.76	21.644	325	-	-	0.97 ± 0.3 ^f^	0.22 ± 0.01 ^e^
Apigenin	0.12	0.35	22.639	325	0.38 ± 0.01 ^b^	8.65 ± 0.012 ^a^	3.13 ± 0.015 ^a^	-
Quercetin	0.44	1.34	18.636	368	0.01 ± 0.02 ^g^	-	1.54 ± 0.02 ^d^	-
Kaempferol	0.09	0.28	21.635	368	-	-	-	-
Cumulative					2.15 ± 0.02	12.63 ± 0.01	14.41 ± 0.02	4.45 ± 0.02

NH: n-hexane, EA: ethyl acetate, M: methanol, Aq: aqueous extract, RT: retention time, λ: wavelength, µg/mg of sample: microgram of polyphenols per milligram of the sample, means with different superscript letters in the column are significantly (*p* < 0.05) different from one another, -: not detected.

#### A. brevifolia Extracts Possess Mild to Moderate Antibacterial Activity

Next, the antibacterial activity of *A. brevifolia* extracts (100 µg/disc) was established by using the disc diffusion method. All extracts showed mild to moderate growth inhibition of selected clinical isolates (Table 4) as compared to ciprofloxacin (10 µg/disc). *A. brevifolia* M extract showed ZOI of 11 mm against cefixime-resistant *E. coli*, *P. aeruginosa*, and MRSA. Likewise, *A. brevifolia* EA extract exhibited a maximum of 12 mm ZOI against cefixime-resistant *E. coli* and MRSA. Lastly, noteworthy growth inhibition was observed in cefixime-resistant *E. coli* in the presence of EA extract (ZOI 12 ± 0.7 mm).

Subsequent evaluation of MIC (Table 5) using the broth dilution method endorsed the results for the initial antibacterial activity. *A. brevifolia* Aq extract was the least active with the highest MIC value of 100.2 µg/mL against cefixime-resistant *P. aeruginosa* and MRSA. The EA extract demonstrated MIC values of 66.3 µg/mL against cefixime-resistant *P. aeruginosa* and 67.9 µg/mL against *E. coli*. Similar MIC values (79 and 82 µg/mL) were obtained in the case of *S. hemolyticus* and MRSA when treated with *A. brevifolia* M extract. Further evaluation of MIC for cefixime validated its resistance profile against selected clinical isolates. Cefixime exhibited an MIC of 20–40 µg/mL against selected clinical isolates, which was higher than the CLSI set value of 0.25 µg/mL [19].

### 2.4. A. brevifolia Ethyl Acetate Extract Showed Total Synergism with Cefixime

The checkerboard method was used to determine the antibacterial efficacy of *A. brevifolia* extracts in combination with cefixime. For each sample, two-fold serial dilutions starting from the MICs were used, where cefixime was diluted vertically while extracts were diluted horizontally in a 96-well plate. Treatment of all clinical isolates with the combination of cefixime and *A. brevifolia* EA extract showed a three- to five-fold reduction (Table 6) in MIC values of the extract. Interestingly, the MIC of cefixime declined four- to ten-fold in the presence of *A. brevifolia* EA extract. This was supported by the fractional inhibitory concentration index (FICI) values, which were ≤0.5, indicating total synergism between the two samples.

Similarly, *A. brevifolia* M extract enhanced the susceptibility of *E. coli* to cefixime by two-fold, with a FICI value of 0.66, demonstrating partial synergism between the extract and antibiotic. The rest of the extracts also demonstrated partial synergism with cefixime, except for NH extract, which showed no synergistic effect against *E. coli*. It appears that the aqueous extract was least effective in mitigating resistance to cefixime, with no synergistic activity (FICI = 1) against *S. hemolyticus*.

### 2.5. The Effect of the Extracts Alone and in Combination Is Time Dependent

Time–kill kinetic studies were performed to assess whether the effect of *A. brevifolia* extracts alone and in combination with cefixime was time-dependent or concentration-dependent. All clinical isolates were tested at MIC, 2MIC, FICI, and 2FICI values. Overall, clinical isolates treated with the combination of cefixime and extracts at FICI and 2FICI values demonstrated significant growth inhibition throughout the treatment duration as compared to individual treatments. The results were comparable with those of ciprofloxacin (positive control), to which the clinical isolates were susceptible. Bacterial growth in samples treated with extracts alone was much lower as compared to cefixime and DMSO (negative control).

Treatment of resistant *E. coli* with *A. brevifolia* EA extract (Figure 3A) at FICI values demonstrated maximum growth inhibition of 91.8% and 81.7% at 3 h and 9 h, respectively, as compared to 50% and 40.3% inhibition at the same time points with cefixime alone. Likewise, the Aq, EA, and NH extracts at 2FICI values showed growth inhibition of 93%, 83.5%, and 67.3% after 9 h of treatment. Treatment of cefixime-resistant *P. aeruginosa* (Figure 3B) with extracts alone (MIC and 2MIC) or cefixime showed inhibition of bacterial growth till 6 h of treatment. Later, there was an exponential increase in bacterial growth as depicted by the increased absorbance of the samples. The combination of cefixime with *A. brevifolia* EA extract at 2FICI demonstrated 100% and 98.7% inhibition of clinical isolates as compared to 66% and 44.9% inhibition with cefixime alone at 6 h and 9 h of treatment, respectively. Similarly, 2FICI dosing of NH, M, and Aq extracts also showed 100%, 89.5%, and 100% inhibition, respectively, of clinical isolates at 6 h of treatment. Although this declined to 73.4%, 74.7%, and 72.4% inhibition at 9 h for NH, M, and Aq extracts, the values were still higher than for cefixime alone (44.9%), indicating continued synergism between samples.

The succeeding analysis on *S. hemolyticus* (Figure 4B) demonstrated a similar pattern of growth as that of E. coli but with different periods. The 2FICI dosing for all extracts was most effective in inhibiting the growth of cefixime-resistant *S. hemolyticus*, which peaked after 6 h of treatment. There was 67.3%, 80.3%, 84.2%, and 76.3% growth inhibition at 6 h with 2FICI of NH, EA, M, and Aq extracts, respectively, as compared to 54.7% inhibition when using cefixime alone.

There was a drastic reduction in MRSA resistance (Figure 4A) to cefixime with FICI and 2FICI dosing. The growth of MRSA clinical isolates was reduced by >10-fold through the synergistic action of cefixime and *A. brevifolia* extracts. Like all other samples, the effect of FICI and 2FICI peaked at 6 h with percent inhibitions of 79.6%, 100%, 87.2%, and 100% at 2FICI for NH, EA < M, and Aq extracts, respectively. It was much higher than the 6.7% growth inhibition by cefixime alone at 6 h. In short, *A. brevifolia* extracts potentiated the antibacterial effect of cefixime at FICI and 2FICI values with a stationary growth phase after treatment, irrespective of time duration.

### 2.6. A. brevifolia Extracts Reduce Bacterial Protein Content

Disintegration of the cell envelope can be quantified using the leakage of cellular protein as a function of cell death. Protein content in the extracellular medium of treated and untreated bacterial strains was analyzed (Table 6) to understand the underlying cause of the antibacterial effect. Bovine serum albumin was used as a positive control. There was little reduction (5.4%) in protein content after treatment of resistant *S. hemolyticus* with cefixime alone. This increased to 79.8%, 68.4%, 78.4%, and 62.7% reduction in protein content when resistant *S. hemolyticus* was treated with the combination of cefixime and NH, EA, M, and Aq extracts, respectively. This indicates that the extracts and antibiotic synergy can degrade the bacterial protein, making its survival difficult. Furthermore, there was 73.9%, 74.4%, 82.5%, and 75% inhibition of protein content in cefixime-resistant MRSA isolates when treated with the combination of NH, EA < M, and Aq extract, respectively, as compared to cefixime (35% reduction). Similarly, a percent protein reduction of 58.8–80.2% was observed in resistant *P. aeruginosa* and *E. coli* isolates due to the synergistic activity of cefixime and *A. brevifolia* extracts. It is postulated that the extracts can inhibit a bacterial protein that works together with the cell wall synthesis inhibitor cefixime to inhibit the growth of resistant clinical isolates. This seems to be the case considering the results of the protein content estimation.

### 2.7. Hemolytic Analysis

Hemolytic analysis was performed to check whether the drug or compound was toxic to red blood cells causing hemolysis. According to ASTM F756-00 protocols for assessment of the hemolytic properties of samples, substances with hemolysis percentages of >5%, <5%, and <2% are considered hemolytic, slightly hemolytic, and non-hemolytic, respectively [22]. To our surprise, the extracts were hemolytic with >5% hemolysis when used alone. However, their hemolytic potential declined when given in combination with cefixime. All combinations except NH/cefixime (6.45% hemolysis) had values ranging between 1.34% and 5.13% for FICI and 0% and 3.78% for 2FICI. This indicated that the majority of combinations were safe to use with either slight or non-hemolytic character. These results were significantly (*p* < 0.05) lower than for the positive control Triton-X 100 (100% hemolysis).

## 3. Discussion

Antimicrobial resistance has multiple causes, and it is a great concern in health sciences as it causes treatment failures and poor prognoses of infectious diseases. Researchers are investigating multiple compounds from natural and synthetic sources that can work either alone or in combination with standard antibiotics to eliminate resistant infections [23]. Phytoconstituents of various medicinal plants have proven antibacterial activity against standard antibiotic-susceptible and -resistant bacteria [24,25,26,27]. Plants act as the greatest apothecary and a potential source of treatments for multiple diseases like arthritis, cancer, diabetes mellitus, and oxidative stress disorder [28,29]. Considering the beneficial aspects of medicinal plants, in the current work, we established the antibacterial activity of *Artemisia brevifolia* extracts in resistant clinical bacterial isolates. We also verified the synergistic interaction between cefixime and *A. brevifolia* extracts that potentiates the antibacterial activity against cefixime-resistant clinical isolates.

*A. brevifolia* extracts were prepared in different solvents to yield variable phytoconstituents based on polarity, as follows:Extraction can be a potentially rate-limiting step when preparing samples for screening bioactive compounds of interest. The efficiency of this step is affected by many factors such as solvent polarity, extraction method, physical characteristics size of sample particles, and period of extraction. These contingency factors were addressed first by selecting four solvents for extraction, n-hexane, methanol, ethyl acetate, and water, depicting variable polarity.Second, samples were macerated for 72 h, ensuring sufficient time for solvents to penetrate the fine particles of powdered plant.Third, maceration was combined with periodic sonication, aiding the diffusion of solvent and extraction of phytoconstituents from powdered plant material. A significant extraction yield (6.1%) was obtained with Aq extract, which indicated the presence of more polar content.

The oligosaccharides, sugars, and resins often solubilize better in distilled water compared to other solvents, contributing to overall extraction yield [30]. Although extraction yield depends on the polarity of phytoconstituents and solvent used for extraction, the maximum yield does not dictate the medicinal value since it is directed by the chemical composition and inherent nature of phytochemicals [31].

Preliminary phytochemical analysis of *A. brevifolia* extracts showed the presence of alkaloids, glycosides, tannins, and terpenoids in different extracts. Alkaloids can inhibit bacterial growth by altering membrane permeability, inhibiting nucleic acid synthesis, and disrupting cell division [32]. In addition, research showed that glycosides such as glycyrrhizin have promising antibacterial effects due to inhibition of RNA synthesis in bacteria [33]. Furthermore, tannins exhibit antidiarrheal, antibacterial, antiviral, antitussive, antitumor, and wound-healing activities [34]. Previously reported studies showed that saponins have detergent-like activity and an antibacterial effect by increasing bacterial cell wall permeability [21]. The presence of these phytoconstituents in *A. brevifolia* extracts can be responsible for the subsequent antibacterial activity of the plant.

Bacteria have started to develop resistance against commonly used antibiotics. Resistant species of *S. aureus*, *S. hemolyticus*, *E. coli*, etc., have been recognized in various clinical settings as causing frequent infections and prolonged duration of the infectious diseases [35,36,37].

Medicinal plants are being investigated to optimize therapy for resistant infections. In the current study, the susceptibility of bacterial clinical isolates to selected antibiotics and *A. brevifolia* extracts was assessed to determine the resistance profile and antibacterial capacity of the samples. The disc diffusion method creates zones of inhibition around the sample-impregnated discs, signifying the antibacterial activity. The greater the size of ZOI, the higher the susceptibility of microorganisms to test samples. Results revealed all the extracts demonstrated mild to moderate antibacterial activity, whereby EA extract was more active against MRSA and R. *S. hemolyticus.* On the contrary, NH and M extracts were active against R. *E. coli* and R. *Pseudomonas aeruginosa*, respectively.

Hydroxylated phenols and phenolic compounds are found to be toxic to many microorganisms. The toxicity of phenols to microorganisms depends on the level of hydroxylation, where higher hydroxylation levels are more toxic to the microorganisms [11]. Genus *Artemisia* is rich in several essential metabolites such as glucosinolates, saponins, cyanogenic glycosides, tannins, unsaturated lactones, phenols, and flavonoids. These phytochemicals are used to treat multiple ailments such as malaria, bacterial infection, cancer, and inflammation [38]. In this study, the antibacterial activity of extracts might have been due to hydroxylated phenols present in this plant, which were quantified through HPLC (emodin, luteolin vanillic acid, syringic acid, gallic acid, coumarins, flavonoids, and flavones) [39].

Furthermore, the susceptibility testing revealed that all clinical isolates were resistant to cefixime. Hence, clinical isolates were treated with cefixime in combination with *A. brevifolia* extracts, to observe the possibility of synergism. The “one drug, one target, one disease” paradigm has become an orthodox pharmaceutical strategy considering the emergence of resistance to even previously potent antibiotics. Currently, a multi-drug-target approach is utilized to augment the efficacy of antibacterial therapy. This pattern shift is dictated by the limited effectiveness, resistance, and side effects of monotherapy [40]. The prime advantage of plant-based drugs is that they can be safe, easily affordable, have minimal or no side effects, and have multiple biological targets. A combination of standard antibiotics with plant-based drugs can provide better synergism with the least side effects, particularly against resistant infections. Synergism can decrease the MIC of many marketed antibiotics in the presence of plant extracts. Research shows that polyphenols decrease beta-lactam resistance while flavonoids, diterpenes, and triterpenes have resistance-modulating abilities on many contemporary antibiotics [28,41,42,43].

Researchers have believed that generally some mechanisms that can cause this interaction are inhibition of the sequence of biochemical paths, membranotropic agent usage to enhance the diffusion of other antibacterial drugs, inhibition of enzymes that protect microorganisms, and a membrane-active agent used in combination [44].

In the current study, the checkerboard method was used to determine the synergistic interaction between cefixime and *A. brevifolia* extracts. Previous research outlined that if the MIC of the extract and antibiotic decreases by four-fold, then the combination is known as synergistic, while if MIC of the first test sample decreases by four-fold and the other by two-fold, then the interaction is known as partial synergistic [45,46]. Treatment of all clinical isolates with a combination of cefixime and *A. brevifolia* EA extract showed a four- to ten-fold reduction in MIC values of cefixime. This was reinforced by the FICI values, which were ≤0.5, indicating total synergism between the two samples. Likewise, *A. brevifolia* M extract enhanced the susceptibility of *E. coli* to cefixime by two-fold, with a FICI value of 0.66, demonstrating partial synergism. The limitation of the checkerboard method is that more resources are used to test antibacterial combinations and more than one antimicrobial cannot be checked at a single time [45]. However, it provided evidence that the extract–cefixime synergism successfully inhibited the growth of both Gram-positive and Gram-negative clinical isolates used in this study, showing a broad spectrum of activity. It is possible that *A. brevifolia* extracts either inhibited the sequence of biochemical paths, enhanced diffusion of cefixime, inhibited protein synthesis of bacteria, or inhibited degradation of antibiotics [44]. Future work is planned to assess the mechanism of synergistic interaction observed in this study.

Next, we determined using time–kill kinetic studies whether the interaction between *A. brevifolia* extracts and cefixime was bactericidal or bacteriostatic. Jacqueline et al. described how time–kill kinetic studies are used to determine the bactericidal effect, which may be dependent on time in place of concentration [47]. It was observed that there was significant inhibition of *E. coli* growth when treated with the combination of *A. brevifolia* extracts and cefixime (FICI, 2FICI) as compared to *A. brevifolia* extracts alone (MIC, 2MIC). DMSO (negative control) did not interfere with the results, with constant exponential growth for all isolates. Cefixime-resistant clinical isolates showed initial growth from 0 to 3 h after treatment with MIC and 2MIC values of the extracts. Then, it gradually started to decline after 3 h of treatment. On the other hand, the growth of E. coli started to decline after 6 h of treatment with *A. brevifolia* NH extract. Samples at their FICI and 2FICI values obstructed the clinical isolates in their stationary phase of the growth curve throughout the treatment duration.

The trend of resistant *S. hemolytic* growth was the log phase, partial death phase, and again log phase for the durations of 0–3, 3–6, and 6–9 h, respectively, when treated at MIC and 2MIC values. On the contrary, clinical isolates treated at 2FICI more or less remained in the stationary phase, with no significant overall growth. Although isolates treated with FICI also displayed greater growth inhibition of cefixime-resistant *S. hemolyticus* as compared to cefixime or extracts alone, an increase in the bacterial growth was observed after 9 h of treatment. Exposure of cefixime-resistant *P. aeruginosa* to MIC and 2MIC of extracts or cefixime alone displayed inhibition of clinical isolates till 6 h of treatment. Later, there was an exponential increase in bacterial growth, as represented by the amplified absorbance of the samples. On the contrary, growth inhibition after treatment with the combination of cefixime and extracts (FICI, 2FICI) was more pronounced as compared to lone treatments. Moreover, *A. brevifolia* M extract imparted better synergism against cefixime-resistant *P. aeruginosa* as compared to other extracts by keeping the clinical isolates in a stationary phase.

The MRSA clinical isolate that was used in the present study showed noteworthy resistance against cefixime with exponential growth. In contrast, there was exponential bacterial growth in the first 3 h of treatment with ciprofloxacin (positive control) but it drastically declined later than 6 h of treatment. Likewise, all *A. brevifolia* (MIC; 2MIC) extracts significantly halted MRSA growth. The competition between survival and growth inhibition of bacteria generated slight variability in percent inhibition values at different time points. However, overall, it could be seen that the combination of extract and cefixime augmented the activity of cefixime against clinical isolates.

In the current study, the combination of cefixime with plant extract against all bacterial isolates showed synergistic interaction when calculating the FICI index in the checkerboard method, while time–kill kinetic curves showed additive interaction. This same pattern of interaction was reported previously for *Helichrysum pedunculatum* plant methanolic extracts when given in synergy with antibiotics against *Staphylococcus aureus* [41]. Likewise, the interactions of antibiotic drugs with acetone extract of seeds of *Garcinia kola* [48] and *Thymus vulgaris* were also reported as synergistic/additive when time–kill kinetic assays were performed [49].

Antibacterial drugs inhibit bacterial growth or kill bacteria by targeting proteins, cell walls, cell membranes, and nucleic acid synthesis [50]. We assessed *A. brevifolia* and cefixime synergy by analyzing the protein content of the medium in clinical isolates. Cefixime disrupts cell wall integrity [51], and its combination with *A. brevifolia* extract significantly (*p* < 0.05) reduced viable proteins. Cefixime’s impact on protein content is limited due to its role in cell wall inhibition. However, combined treatment likely curbed protein synthesis or induced apoptosis, lowering protein content, suggesting *A. brevifolia’s* influence on bacterial protein synthesis, potentially through tRNA release inhibition, peptide bond synthesis, or initiation of complex formation [50]. Similar effects were observed for *Cymbopogon khasianus* on resistant clinical isolates [3].

HPLC quantification was performed using 14 polyphenol standards, and it confirmed the flavonoids and polyphenols in *A. brevifolia.* Some of the phytochemicals present in *A. brevifolia* were vanillic acid, gallic acid, caffeic acid, and syringic acid [39]. A literature review showed that vanillic acid ruptures the cell membrane and inhibits cell growth [52], while gallic acid inhibits biofilm production and disrupts the bacterial cell membrane [53]. Caffeic acid acts by inhibiting the RNA polymerase enzyme, and syringic acid is an ATP synthesis inhibitor [11]. All these phytochemicals have been quantified in the subject plant. Therefore, it can be hypothesized that when these polyphenols combined with cefixime resulted in an additive mechanism of action, as suggested in the literature, phytochemicals may have acted upon the cell wall integrity [54]. It has been also testified that some plant chemical compounds inhibit bacterial growth or improve the effect of antibacterial drugs by acting in the same site as peptidoglycan [55].

In the present study, we performed a toxicity analysis as a part of the efficacy study. For this purpose, a hemolytic assay was performed that gives information regarding the cytotoxicity of samples on blood. This model is frequently used because of the easy availability and isolation protocols of red blood cells. Moreover, the membrane physiology of red blood cells is similar to the membranes of other cells present in the body [56]. *A. brevifolia* extracts in combination with cefixime displayed slightly hemolytic or non-hemolytic character; this was different from *A. brevifolia* NH and M extracts alone, which presented hemolytic character. The EA extracts were found to be safer to use in humans as a component of antibacterial therapy. Yet, in vivo toxicity studies must be conducted to determine detailed toxicity versus efficacy profiles.

The strengths of this study are evident in the significant findings that highlight the potential of *A. brevifolia* as a potent antibacterial agent. The identification of substantial minimum inhibitory concentrations in both ethyl acetate and methanolic extracts against resistant bacterial strains, including *Staphylococcus haemolyticus*, methicillin-resistant *Staphylococcus aureus*, *Escherichia coli*, and *Pseudomonas aeruginosa*, underscores the broad-spectrum antibacterial activity of these extracts. Moreover, the observation of total synergistic effects when the ethyl acetate extract was used in combination against all bacterial strains is particularly promising. The study’s assessment of safety, as indicated by negligible hemolysis of red blood cells, adds to its credibility.

However, it is important to acknowledge the limitations of this research. While the in vitro findings are encouraging, the transition to in vivo studies is crucial to determine the real efficacy and potential toxicity effects when these extracts are used in living organisms. Clinical controlled trials are necessary to validate the therapeutic potential of these extracts in treating infectious diseases, and further research should aim to increase the variety of drugs, expand the number of clinical isolates, and identify the specific compounds within the plant extracts responsible for their antibacterial properties. Additionally, a deeper exploration into the mechanism of action and the formulation of pharmacological agents based on these extracts would enhance the practical application of this promising research.

## 4. Materials and Methods

### 4.1. Materials

#### 4.1.1. Chemicals and reagents

Methanol (M), ethyl acetate (EA), n-hexane [49], acetonitrile (ACN), acetic acid (AA), sulfuric acid, nutrient agar, nutrient broth, fetal bovine serum, and Giemsa stain were purchased from Merck (Darmstadt, Germany). Dimethyl sulfoxide (DMSO) was purchased from Duksan Pure Chemicals (Ansan-si, Republic of Korea). All other chemicals and drugs were purchased from Sigma-Aldrich (Darmstadt, Germany) [51] unless otherwise stated.

#### 4.1.2. Cultures

Four resistant bacterial clinical isolates that were maintained in the laboratory were used to evaluate the antimicrobial potential of extracts. These were resistant (R.) *Staphylococcus* (*S.) hemolyticus* (MIC-101), R. *Escherichia coli* [57] (MIC-102), R. *Pseudomonas* (*P.) aeruginosa* (MIC-103), and methicillin resistant *S. aureus* (MRSA; MIC-104).

### 4.2. Methods

#### 4.2.1. Preparation of Extracts

*A. brevifolia* was collected in August 2018 from Hunza Valley, Baltistan by Dr. Ihsan ul Haq, Associate Professor, Department of Pharmacy, Quaid-i-Azam University, Islamabad. It was identified by Dr Sher Wali Khan, Karakoram International University, Gilgit Baltistan. The specimen was submitted (Herbarium No# PHM 512) to the Herbarium of Medicinal Plant, Quaid-i-Azam University, Islamabad, Pakistan. The collection and investigations of *A. brevifolia* were supported in part by the Indigenous Fellowship of HEC provided to the first author (520-142973-2MD6-130 (50093185)). About 12 kg of plant material was washed, shade dried (3 weeks), pulverized to a coarse powder, and stored in an airtight container. Successive maceration aided with ultra-sonication was used to extract plant material as previously reported [31]. Dry powder was macerated with four analytical-grade solvents (non-polar to polar) including NH, EA, M, and distilled water (Aq) at a ratio of 1:4 (powder:solvent) for 72 h at 25 °C with 10 min of sonication each day. After 3 days, extracts were filtered and concentrated using a reduced-pressure rotary evaporator (Ribby, UK) at 45 °C. Plant extracts were collected in labeled containers and stored at −80 °C temperature for further testing. The dried extracts were weighed to calculate the percentage extract recovery using the formula:(1)$$\%\mathrm{age}\;\mathrm{Extract}\;\mathrm{Recovery}=\frac{Total\;weight\;of\;extract\;after\;drying}{Total\;weight\;of\;plant\;powder}\times 100,$$

#### 4.2.2. Total Phenolic Content

The protocol used to determine total phenolic content was given by [31] as was subject to few modifications. In our work, 20 µL of each test sample (4 mg/mL) was taken and poured into a 96-well plate followed by the addition of 90 µL Folin–Ciocalteu reagent. The plate was incubated for 5 min, and then sodium carbonate was added to the reaction mixture. After that, the absorbance of the plate was taken at 630 nm using a microplate reader (BioTek; Shoreline, USA). DMSO was used as negative while gallic acid was used as positive control. The assay was carried out in triplicate and results were given as mg gallic acid equivalent per gram dry weight.

#### 4.2.3. Glycoside, Alkaloidal, Tannin, Saponin, and Terpenoid Contents

Three types of glycosides (cardiac, anthraquinone, and coumarin) were determined using the method given by Shaikh et al. [58] with minor modifications. Cardiac glycosides were confirmed through Keller Killani, Salkowski, and Baljet tests. Anthraquinone glycosides were estimated via borax and modified Bontrager’s tests. On the other hand, the sodium-hydroxide-mediated fluorescence method indicated the presence of coumarin glycosides.

The alkaloidal content of extracts was determined using Wagner’s, tannic acid, and Dragendroff’s reagents, as given by [59].

Tannins in *A. brevifolia* extracts were detected using the protocol given by [59]. Ferric chloride and gelatin solution were used for this purpose.

Saponin content in *A. brevifolia* extracts was evaluated via foam formation with or without olive oil, as previously described by [60].

Phytochemical analysis for the presence of terpenoids was performed using the method given by [61], which uses chloroform and sulfuric acid to precipitate reddish brown terpenoids.

#### 4.2.4. Antimicrobial Evaluation: Preliminary Resistance Profiling of Antibiotics

Initially, antibiotics were tested against clinical isolates by using the disc diffusion method. Stock solutions (4 mg/mL) of antibiotics (cefixime, ciprofloxacin, doxycycline, lincomycin, and clarithromycin) were prepared in DMSO. Agar media was poured on plates and sterile discs loaded with 5 µL of antibiotics were placed on the plates and incubated for 24 h at 37 °C. The zone of inhibition (ZOI) around each disc was measured using a vernier caliper. The assay was carried out in triplicate. The antibiotic with little or no ZOI was selected for further studies. Minimum inhibitory concentrations [21] of antibiotics were determined by using the micro broth dilution protocol reported by [62] with little modification.

#### 4.2.5. RP-HPLC Analysis

##### LOD and LOQ Determination

LOD and LOQ for HPLC analysis were determined. LOD stands for the limit of detection, which is the lowest concentration of sample that can be detected in HPLC, while LOQ stands for the limit of quantification, which is the minimum concentration of sample that can be quantified using HPLC. These two parameters are determined using the following formulae:(2)$$\mathrm{LOD}=3.3\times \frac{SD}{S}$$
(3)$$\mathrm{LOQ}=10\times \frac{SD}{S}$$
where SD = standard deviation of regression and S = slope of the calibration curve.

##### Analysis of Polyphenols

To identify and measure the number of polyphenols present in *A. brevifolia* crude extracts, RP-HPLC was utilized according to the standard protocol with minor modifications [20,39]. A zorbex-C8 analytical column (5 µm; 4.6 × 250 mm) connected with a diode array detector (DAD) was supplied with an HPLC system (Shimadzu; Kyoto, Japan). A binary gradient system with mobile phase A (methanol: water: acetic acid: acetonitrile in 10:85:1:5) and mobile phase B (acetonitrile: methanol: acetic acid in 40:60:1) was used to accomplish the polyphenol detection. The column was injected with 50 µL of sample solution prepared in methanol, and the flow rate was adjusted at 1.2 mL/min. The gradient volume of mobile phase B was changed from 0% to 75% in the first 0–30 min, to 75–100% in the next 30–31 min, then to 100% in 31 to 35 min and, lastly, to 0% in the final 36 to 40 min. The column was reconditioned before injecting a new sample. The concentration of each standard was 50 µg/mL in methanol, while extracts were prepared at 10 mg/mL in methanol. The quantification of polyphenols was performed by comparing the UV-Vis spectra and retention times of chromatographic peaks to reference standards; polyphenols were identified at 257 nm for vanillic acid and rutin, 279 nm for apocynin, coumaric acid, catechin, syringic acid, and gallic acid, 325 nm for apigenin, caffeic acid, ferulic acid, gentisic acid, and luteolin, and 368 nm for quercetin and kaempferol. The results were quantified in terms of µg/ mg extract (µg/mgE).

#### 4.2.6. Antibacterial Assay

The antibacterial potential of *A. brevifolia* extracts was determined by using the disc diffusion method [62] as described above. A bacterial culture at a seeding density of 1 × 10^6^ CFU/mL was used to make bacterial spreads on nutrient agar plates. About 5 µL of each test sample (cefixime/ciprofloxacin, 20 µg/disc; extracts, 100 µg/disc) was poured on sterile filter discs, and then placed on a nutrient agar plate and incubated (24 h at 37 °C). Ciprofloxacin was used as a positive control, whereas DMSO served as a negative control. After 24 h, ZOI was calculated by using a vernier caliper. The assay was run in triplicate.

Next, samples that showed ZOI ≥ 12 mm were further tested to determine the MIC by using the micro broth dilution method [62]. A bacterial inoculum was prepared by adjusting the seeding density at 5 × 10^4^ CFU/mL. Three-fold serial dilutions of extracts (100, 33.3, 11.1, and 3.34 µg/mL) and antibiotics (10, 3.33, 1.11, and 0.334 µg/mL) were prepared in nutrient broth. About 5 µL of the test sample and 195 µL of inoculum were mixed in each well of a 96-well plate. The plate was then incubated at 37 °C for 24 h, and absorbance was measured at 600 nm after 30 min (0 h reading) and 24 h of incubation in a microplate reader (BioTek; Shoreline, DC, USA).

The checkerboard method was used to determine that the potential synergistic interaction between antibiotic and *A. brevifolia* extracts [63]. Two-fold serial dilutions of the sample were prepared such that the antibiotic was diluted vertically while extracts were diluted horizontally in a 96-well plate. An aliquot of 5 µL of sample was poured in each well containing 2.5 µL of extract and 2.5 µL of antibiotic, followed by 195 µL of inoculum (density 4 × 10^4^ colony-forming unit (CFU)/mL) in each well. The plates were then incubated at 37 °C for 24 h, and FICI values were determined. Absorbance was taken at 0 h and after 24 h for further calculations:(4)$$\mathrm{FICI}=\frac{{MICI}_{B}^{A}}{{MIC}_{A}}+\frac{{MICI}_{A}^{B}}{{MIC}_{B}},$$
where MICI_A/B_ = MIC of compound A in combination with compound B, MIC_A_ = MIC of compound A, MICI_B/A_ = MIC of compound B in combination with compound A, and MIC_B_ = MIC of compound B.

The interaction between antibiotic and extract was considered “total synergism” or “partial synergism” at FICI ≤ 0.5 and 0.5 < FICI ≤ 0.75, respectively. If FICI was ≤1 or between 1 and 4, then the interaction was termed “Indifference” or “No effect”. If the FICI value was more than 4, then it was “Antagonism” [63].

Time–kill kinetics was performed using the protocol described previously [3,64] with few modifications. Resistant bacterial strains were grown to the mid-logarithmic phase. Bacterial cells were diluted up to 10^4^ CFU/mL. This bacterial suspension was then incubated with MIC, 2MIC, FICI, and 2FICI concentrations of extracts alone and in combination with selected antibiotics. Readings were taken at time intervals of 0, 3, 6, and 9 h after incubation. Results were measured using a UV spectrophotometer (Sigma Aldrich; Darmstadt, Germany) at 600 nm.

Bacteria were grown up to the mid-logarithm phase, as described previously, and were treated with MIC, 2MIC, FICI, and 2FICI of extract alone and in combination with selected antibiotic [65]. Protein content estimation of clinical isolates was conducted using the Bradford reagent to check the possible mechanism of action of bacterial growth inhibition. After incubation of samples clinically isolated for 24 h, 5 µL of the reaction mixture was mixed with 195 µL of Bradford reagent and incubated for 5 min at room temperature with constant sonication in triplicate. Then, absorbance was measured at 595 nm. The protein content of samples was calculated using the formula:(5)$$\mathrm{Absorbance}\;\mathrm{of}\;\mathrm{unknown}\;\mathrm{sample}\;(x)=\frac{Absorbance-b\;}{m},$$

Phosphate buffer was used as a diluent in this assay. The negative control, positive control, and blank were constituted by the clinical isolate inoculum, bovine albumin serum (0–50 µg/mL), and distilled water, respectively.

#### 4.2.7. Hemolytic Assay

Hemolytic evaluation was performed using freshly drawn human blood following the guidelines given by the ethical committee of Quaid-i-Azam University, Islamabad, Pakistan. Bioethical approval was given by the Ethical Committee of the university (BEC-FBS-QAU2021-261 dated 2 March 2021). Informed consent was also obtained from the volunteer to draw blood samples. Blood was collected and centrifuged at 13,000 rpm to separate red blood cells (RBCs). After that, RBCs were washed thrice with normal saline and re-suspended in phosphate buffer to form a 5% solution. The RBC suspension was incubated with samples at 37 °C for 30 min in Eppendorf (2 mL) tubes. Afterward, it was centrifuged at 2000 rpm for 20 min and the supernatant was separated. Triton X-100 (0.1%) was used as the positive control, whereas phosphate buffer served as the negative control [3]. The absorbance of the supernatant was measured at 360 nm in a microplate reader (BioTek; Shoreline USA). Hemolysis was calculated using the following formula:(6)$$\%\mathrm{age}\;\mathrm{Hemolysis}=\frac{Absorbance\;of\;sample-Absorbance\;of\;Negative}{Absorbance\;of\;positive-Absorbance\;of\;Negative}\times 100,$$

### 4.3. Statistical Evaluation

Statistical analysis of the experimental results was conducted utilizing GraphPad Prism 5 (version 5.00 for Windows, San Diego, CA, USA). The presented data included mean values accompanied by either the standard error of the mean (SEM) or the standard deviations (SDs) of individual replicates. Additionally, for each experiment, the analysis software employed and the number of observations were specified.

## 5. Conclusions

In this research, all *A. brevifolia* extracts that were examined exhibited noteworthy antibacterial activity. Particularly, the ethyl acetate extract demonstrated significant activity against resistant strains of *Staphylococcus hemolyticus* and methicillin-resistant *Staphylococcus aureus*. Additionally, the methanol extract displayed notable inhibition of growth in resistant strains of *Escherichia coli* and *Pseudomonas aeruginosa*.

Furthermore, the ethyl acetate extract exhibited complete synergism when combined with cefixime against all cefixime-resistant clinical isolates. This synergistic effect was observed to be time-dependent, with the maximum bactericidal activity occurring at 2FICI concentrations after 6 and 9 h of treatment, varying among different bacterial strains.

Notably, our study revealed a significant reduction in the protein content of bacterial samples, suggesting potential mechanisms of action. This reduction could be attributed to the combined effects of protein synthesis inhibition or protein degradation induced by the *A. brevifolia*/cefixime combination, along with the disruption of cell wall integrity caused by cefixime.

In light of these findings, our research strongly advocates for further comprehensive investigations into the application of *A. brevifolia* extracts in synergy with cefixime as a potential strategy for combating bacterial infections that have developed resistance. These promising results warrant in-depth exploration and consideration for the development of novel treatments for drug-resistant bacterial infections.

## Figures and Tables

**Figure 1 antibiotics-12-01553-f001:**
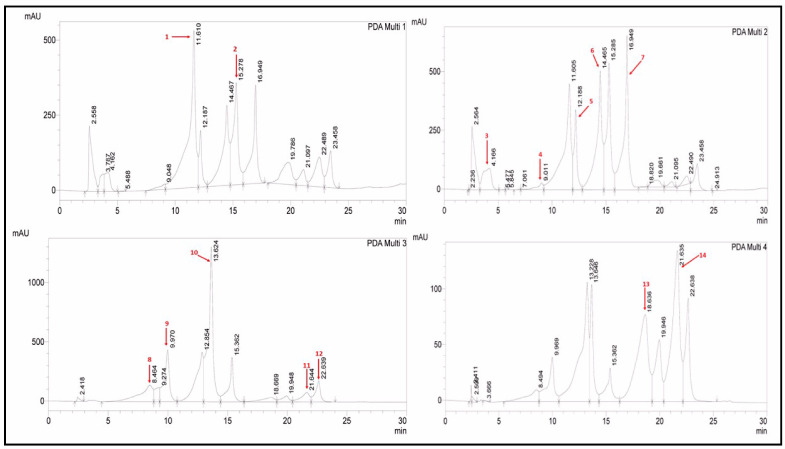
RP-HPLC chromatograms of 14 standards showing the presence of polyphenols and derivatives. 1: vanillic acid, 2: rutin, 3: gallic acid, 4: catechin, 5: syringic acid, 6: apocynin, 7: coumaric acid, 8: gentisic acid, 9: caffeic acid, 10: ferulic acid, 11: luteolin, 12: apigenin, 13: quercetin, 14: kaempferol.

**Figure 2 antibiotics-12-01553-f002:**
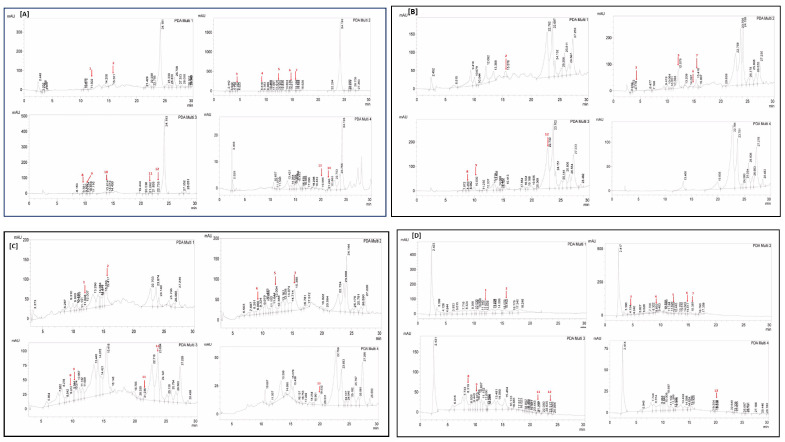
RP-HPLC chromatograms of *A. brevifolia* extracts for polyphenols and derivatives. Chromatograms of (**A**) n-hexane extract, (**B**) ethyl acetate extract, (**C**) methanol extract, and (**D**) aqueous extract showing the presence of polyphenols and derivatives. 1: vanillic acid, 2: rutin, 3: gallic acid, 4: catechin, 5: syringic acid, 6: apocynin, 7: coumaric acid, 8: gentisic acid, 9: caffeic acid, 10: ferulic acid, 11: luteolin, 12: apigenin, 13: quercetin, 14: kaempferol.

**Figure 3 antibiotics-12-01553-f003:**
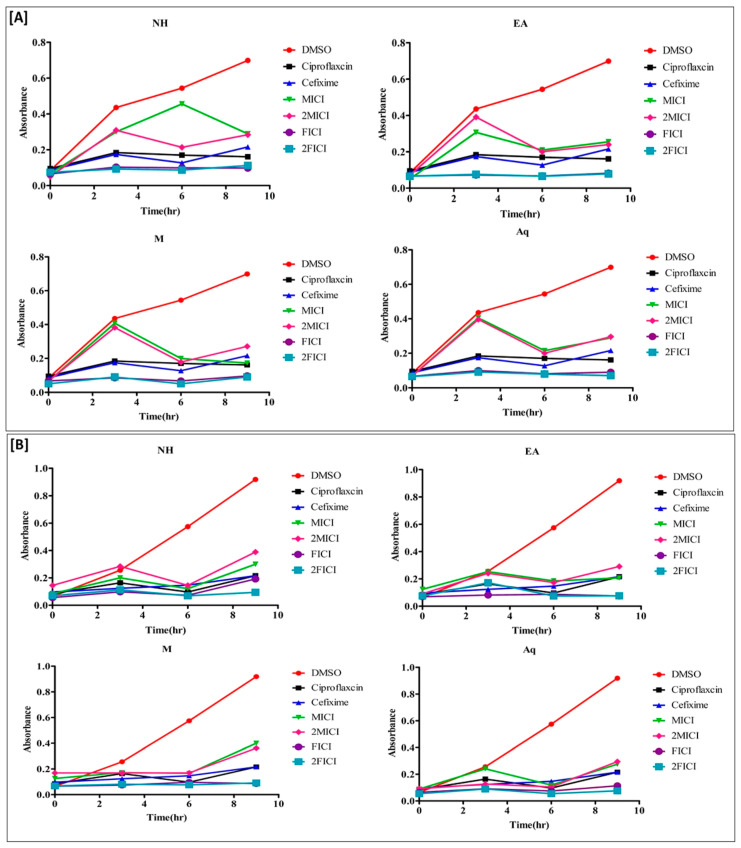
Time–kill kinetics curves of *Artemisia brevifolia*, cefixime, and their combination against cefixime -resistant Gram-negative bacterial strains (**A**) *E. coli* and (**B**) R. *P. aeruginosa.* The count of dead cells was monitored for 0, 3, 6, and 9 h. The color of a line indicates the concentration of the treatment used in the experiment: red, untreated control; black, positive control; blue, 1X MIC of cefixime; green, 1X MIC; pink, 2X MIC; purple, 1X FICI; and yellow, 2X FICI.

**Figure 4 antibiotics-12-01553-f004:**
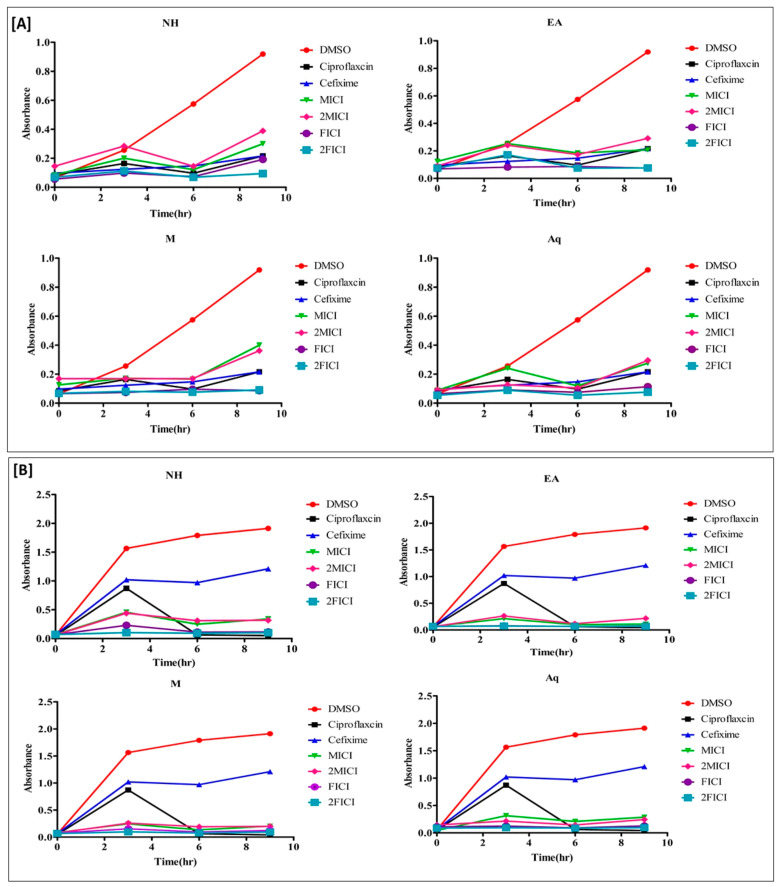
Time–kill kinetics curves of *Artemisia brevifolia*, cefixime, and their combination against cefixime-resistant Gram-positive bacterial strains (**A**) R. *S. hemolyticus* and (**B**) MRSA. The count of dead cells was monitored for 0, 3, 6, and 9 h. The color of a line indicates the concentration of the treatment used in the experiment: red, untreated control; black, positive control; blue, 1X MIC of cefixime; green, 1X MIC; pink, 2X MIC; purple, 1X FICI; and yellow, 2X FICI.

**Table 1 antibiotics-12-01553-t001:** Phytochemical analysis of *A. brevifolia*.

Extract	TFC (µgQE/mgE)	TPC (µgGAE/mgE)
NH	1.72 ± 0.02 ^d^	1.52 ± 0.029 ^d^
EA	2.82 ± 0.3 ^c^	5.2 ± 0.04 ^c^
M	9.93 ± 0.5 ^a^	15.98 ± 0.1 ^a^
Aq	7.65 ± 0.025 ^b^	11.25 ± 0.3 ^b^

NH: n-hexane, EA: ethyl acetate, M: methanol, Aq: aqueous extract, TFC: total flavonoid content, TPC: total phenolic content, µgQE/mgE: microgram quercetin equivalent per milligram of extract, µgGAE/mgE: microgram gallic acid equivalent per milligram of extract. Means with different superscript (^a–d^) letters in the column are significantly (*p* < 0.05) different from one another.

**Table 2 antibiotics-12-01553-t002:** Secondary metabolite screening of *A. brevifolia*.

Extract	Glycosides			Alkaloids	Saponins	Tannins	Terpenoids
	Cardiac	Anthraquinone	Coumarin				
NH	+	−	−	+	−	−	+
EA	+	+	−	−	−	−	+
M	+	+	+	+	−	+	−
Aq	+	+	+	+	+	+	−

NH: n-hexane, EA: ethyl acetate, M: methanol, Aq: aqueous extract, “+” indicates presence; “−” indicates absence.

**Table 4 antibiotics-12-01553-t004:** Antibacterial susceptibility testing of antibiotics from major antibiotic class.

Antibiotic (20 µg/disc)	Antibacterial Activity (ZOI mm ± SD)			
	R. *E. coli*	R. *P. aeruginosa*	R. *S. haemolyticus*	MRSA
Ciprofloxacin	17 ± 0.6 ^c^	16 ± 0.1 ^c^	24 ± 0.4 ^b^	20 ± 0.12 ^c^
Doxycycline	30 ± 0.1 ^a^	25 ± 0.76 ^b^	24 ± 0.01 ^b^	28 ± 0.11 ^b^
Cefixime	-	-	-	-
Clarithromycin	30 ± 0.3 ^a^	35 ± 0.23 ^a^	37 ± 0.1 ^a^	35 ± 0.10 ^a^
Lincomycin	21 ± 0.3 ^b^	25 ± 0.5 ^b^	24 ± 0.1 ^b^	20 ± 0.01 ^c^

ZOI: zone of inhibition, SD: standard deviation, “-” indicates no activity. Means with different superscript (^a–c^) letters in the column are significantly (*p* < 0.05) different from one another.

**Table 5 antibiotics-12-01553-t005:** Minimum inhibitory concentration of *A. brevifolia* extracts and cefixime.

Pathogen	*A. brevifolia* Extracts (µg/mL)				Cefixime (µg/mL)	Ciprofloxacin (µg/mL)
	NH	EA	M	Aq		
R. *P. aeruginosa*	80.9	66.3	66.6	100.2	40	1.11
R. *E. coli*	69.56	84.53	73.5	105.7	40	1.11
R. *S. haemolyticus*	79.17	79.3	79	100.4	40	0.37
MRSA	86.4	67.9	82	100.2	20	3.33

NH: n-hexane, EA: ethyl acetate, M: methanol, Aq: aqueous extract, R. *P. aeruginosa*: resistant *Pseudomonas aeruginosa*, R. *E. coli*: resistant *Escherichia coli*, R. *S. haemolyticus*: resistant *Staphylococcus haemolyticus*, MRSA: methicillin-resistant *Staphylococcus aureus*.

**Table 6 antibiotics-12-01553-t006:** Minimum inhibitory concentration [21] of *A. brevifolia* and cefixime alone and in combination against cefixime-resistant bacterial strains.

Strain		MIC (µg/mL) (Alone)	MIC (µg/mL) Combination	Fold Reduction	Fractional Inhibitory Concentration Index (FICI)	Synergism
R. *P. aeruginosa*	NH	80.9	30	3	0.62	Partial
	Cefixime	40	10	4		
	EA	66.3	20	3	0.36	Total
	Cefixime	40	2.5	16		
	M	66.6	30	2	0.58	Partial
	Cefixime	40	5	8		
	Aq	100.2	50	2	0.75	Partial
	Cefixime	40	10	4		
R. *E. coli*	NH	69.56	40	2	0.83	Indifferent
	Cefixime	40	10	4		
	EA	84.53	20	4	0.36	Total
	Cefixime	40	5	8		
	M	73.5	30	2	0.66	Partial
	Cefixime	40	10	4		
	Aq	105.7	25	4	0.74	Partial
	Cefixime	40	20	2		
R. *S. haemolyticus*	NH	79.17	20	4	0.75	Partial
	Cefixime	40	20	2		
	EA	79.3	15	5	0.44	Total
	Cefixime	40	10	4		
	M	79	20	4	0.75	Partial
	Cefixime	40	20	2		
	Aq	100.4	50	2	1.00	Indifferent
	Cefixime	40	20	2		
MRSA	NH	86.4	40	2	0.71	Partial
	Cefixime	20	5	4		
	EA	67.9	15	5	0.28	Total
	Cefixime	20	1.25	16		
	M	82	40	2	0.74	Partial
	Cefixime	20	5	4		
	Aq	100.2	25	4	0.75	Partial
	Cefixime	20	10	2		

NH: n-hexane, EA: ethyl acetate, M: methanol, Aq: aqueous, MIC: minimum inhibitory concentration.

## Data Availability

Data related to the present research are provided throughout the manuscript and/or are supported by the inserted references.

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
