# Peer review of "Artemisia brevifolia Wall. Ex DC Enhances Cefixime Susceptibility by Reforming Antimicrobial Resistance"

_antibiotics, 2023, doi:10.3390/antibiotics12101553_

Round 1
Reviewer 1 Report
Dear Authors,
I have noticed that a similar paper was already published by your group. I did not see any major different in this manuscript with your published paper. No one should understand journal publication in the form of paper-mills.
Syeda Tayyaba Batool Kazmi, Iffat Naz, Syeda Saniya Zahra, Hamna Nasar, Humaira Fatima, Ayesha Shuja Farooq, Ihsan-ul Haq, Phytochemical analysis and comprehensive evaluation of pharmacological potential of Artemisia brevifolia Wall. ex DC, Saudi Pharmaceutical Journal, Volume 30, Issue 6, 2022, Pages 793-814, ISSN 1319-0164,
Language should be edited.
Author Response
Please see the attached file for the reviewer's comments response.

Reviewer 2 Report
1. Minor editing of the English language is required
2. The novelty of the study should be included in the introduction.
3. How the results are significantly different which should be mentioned in the results section (Table 1, 3... etc.). see https://doi.org/10.1007/s12010-021-03669-8 and maybe cite.
4. LOD, LOQ, etc. are required for HPLC analysis.
5. The "p" value should be in italics.
6. The conclusion part needs to be re-written
Minor editing of the English language is required
Author Response
Please see the attachment for the response to reviewer's comments.

Reviewer 3 Report
Overall, in my opinion, an interesting and clearly presented study. It would be interesting to test the single polyphenols or secondary metabolites in combination with Cefixime to determine the main compounds responsible for the observed effect.
I have no comments on the scientific part, but I'd suggest to the authors to avoid single-sentence paragraphs.
Perhaps the only question I'd like to ask (just a comment by the authors in their response) is the following: giving the composition heterogeneity of the different extracts, how would the authors explain the similar relative effect of the same (alone and in combination) in the time dependent-experiments?
The text is clear, there are only minor mistakes. Some editorial work is required.
Round 2
Reviewer 1 Report
Dear Authors,
Thank you for your clarity about this manuscript than your previous publication. Yes, I agree with your points and suggest to editor to consider this work. However, you should review my following points.
1) There should not be intercept in calibration curve of standard to estimate TPC and TFC. Why did not you follow the equation y=mx? (not, y=mx+c)
2) In your tables 5 and 6, MIC data are same for various microorganisms and extracts. Why?
3) In figure 3 and 4, data presented on Time-kill Kinetics curve should be clear to readers.
Minor editing required.
Author Response
Please see the attachment for author's notes to reviewer
